# Reflection, Refraction, and Hamiltonian Monte Carlo

**Hadi Mohasel Afshar**
Research School of Computer Science
Australian National University
Canberra, ACT 0200
hadi.afshar@anu.edu.au

**Justin Domke**
National ICT Australia (NICTA) &
Australian National University
Canberra, ACT 0200
Justin.Domke@nicta.com.au

## Abstract

Hamiltonian Monte Carlo (HMC) is a successful approach for sampling from continuous densities. However, it has difficulty simulating Hamiltonian dynamics with non-smooth functions, leading to poor performance. This paper is motivated by the behavior of Hamiltonian dynamics in physical systems like optics. We introduce a modification of the Leapfrog discretization of Hamiltonian dynamics on piecewise continuous energies, where intersections of the trajectory with discontinuities are detected, and the momentum is *reflected* or *refracted* to compensate for the change in energy. We prove that this method preserves the correct stationary distribution when boundaries are affine. Experiments show that by reducing the number of rejected samples, this method improves on traditional HMC.

## 1 Introduction

Markov chain Monte Carlo sampling is among the most general methods for probabilistic inference. When the probability distribution is smooth, Hamiltonian Monte Carlo (HMC) (originally called hybrid Monte Carlo [4]) uses the gradient to simulate Hamiltonian dynamics and reduce random walk behavior. This often leads to a rapid exploration of the distribution [7, 2]. HMC has recently become popular in Bayesian statistical inference [13], and is often the algorithm of choice.

Some problems display *piecewise* smoothness, where the density is differentiable except at certain boundaries. Probabilistic models may intrinsically have finite support, being constrained to some region. In Bayesian inference, it might be convenient to state a piecewise prior. More complex and highly piecewise distributions emerge in applications where the distributions are derived from other distributions (e.g. the distribution of the product of two continuous random variables [5]) as well as applications such as preference learning [1], or probabilistic programming [8].

While HMC is motivated by smooth distributions, the inclusion of an acceptance probability means HMC *does* asymptotically sample correctly from piecewise distributions[1]. However, since leapfrog numerical integration of Hamiltonian dynamics (see [9]) relies on the assumption that the corresponding potential energy is smooth, such cases lead to high rejection probabilities, and poor performance. Hence, traditional HMC is rarely used for piecewise distributions.

In physical systems that follow Hamiltonian dynamics [6], a discontinuity in the energy can result in two possible behaviors. If the energy decreases across a discontinuity, or the momentum is large enough to overcome an increase, the system will cross the boundary with an instantaneous change in momentum, known as *refraction* in the context of optics [3]. If the change in energy is too large to be overcome by the momentum, the system will *reflect* off the boundary, again with an instantaneous change in momentum.

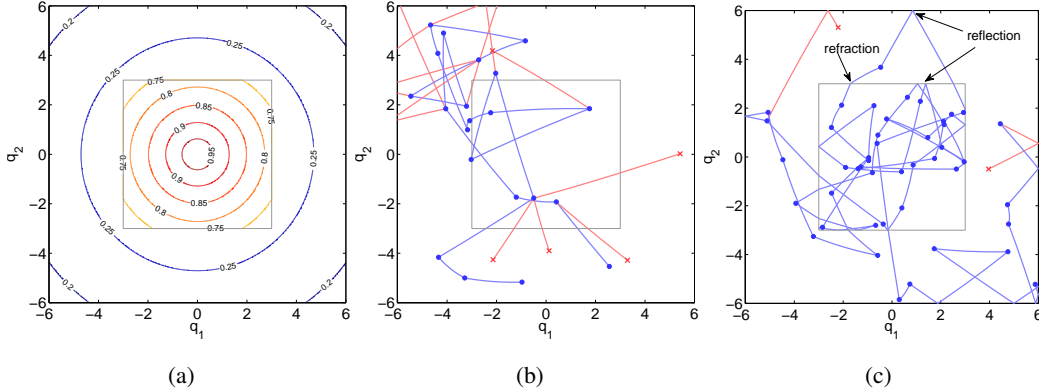

Figure 1: Example trajectories of baseline and reflective HMC. (a) Contours of the target distribution in two dimensions, as defined in Eq. 18. (b) Trajectories of the rejected (red crosses) and accepted (blue dots) proposals using baseline HMC. (c) The same with RHMC. Both use leapfrog parameters $L = 25$ and $\epsilon = 0.1$ In RHMC, the trajectory reflects or refracts on the boundaries of the internal and external polytope boundaries and thus has far fewer rejected samples than HMC, leading to faster mixing in practice. (More examples in supplementary material.)

Recently, Pakman and Paninski [11, 10] proposed methods for HMC-based sampling from piecewise Gaussian distributions by exactly solving the Hamiltonian equations, and accounting for what we refer to as refraction and reflection above. However, since Hamiltonian equations of motion can rarely be solved exactly, the applications of this method are restricted to distributions whose log-density is piecewise quadratic.

In this paper, we generalize this work to arbitrary piecewise continuous distributions, where each region is a polytope, i.e. is determined by a set of affine boundaries. We introduce a modification to the leapfrog numerical simulation of Hamiltonian dynamics, called Reflective Hamiltonian Monte Carlo (RHMC), by incorporating reflection and refraction via detecting the first intersection of a linear trajectory with a boundary. We prove that our method has the correct stationary distribution, where the main technical difficulty is proving volume preservation of our dynamics to establish detailed balance. Numerical experiments confirm that our method is more efficient than baseline HMC, due to having fewer rejected proposal trajectories, particularly in high dimensions. As mentioned, the main advantage of this method over [11] and [10] is that it can be applied to arbitrary piecewise densities, without the need for a closed-form solution to the Hamiltonian dynamics, greatly increasing the scope of applicability.

## 2  Exact Hamiltonian Dynamics

Consider a distribution $P(\mathbf{q}) \propto \exp(-U(\mathbf{q}))$ over $\mathbb{R}^n$, where $U$ is the *potential energy*. HMC [9] is based on considering a joint distribution on momentum and position space $P(\mathbf{q}, \mathbf{p}) \propto \exp(-H(\mathbf{q}, \mathbf{p}))$, where $H(\mathbf{q}, \mathbf{p}) = U(\mathbf{q}) + K(\mathbf{p})$, and $K$ is a quadratic, meaning that $P(\mathbf{p}) \propto \exp(-K(\mathbf{p}))$ is a normal distribution. If one could exactly simulate the dynamics, HMC would proceed by (1) iteratively sampling $\mathbf{p} \sim P(\mathbf{p})$, (2) simulating the Hamiltonian dynamics

$$\frac{\mathrm{d}q_i}{\mathrm{d}t} = \frac{\partial H}{\partial p_i} = p_i \qquad (1) \qquad\qquad \frac{\mathrm{d}p_i}{\mathrm{d}t} = -\frac{\partial H}{\partial q_i} = -\frac{\partial U}{\partial q_i} \qquad (2)$$

for some period of time $\epsilon$, and (3) reversing the final value $\mathbf{p}$. (Only needed for the proof of correctness, since this will be immediately discarded at the start of the next iteration in practice.) Since steps (1) and (2-3) both leave the distribution $P(\mathbf{p}, \mathbf{q})$ invariant, so does a Markov chain that alternates between the two steps. Hence, the dynamics have $P(\mathbf{p}, \mathbf{q})$ as a stationary distribution. Of course, the above differential equations are not well-defined when $U$ has discontinuities, and are typically difficult to solve in closed-form.

# 3 Reflection and Refraction with Exact Hamiltonian Dynamics

Take a potential function $U(\mathbf{q})$ which is differentiable in all points except at some boundaries of partitions. Suppose that, when simulating the Hamiltonian dynamics, $(\mathbf{q}, \mathbf{p})$ evolves over time as in the above equations whenever these equations are differentiable. However, when the state reaches a boundary, decompose the momentum vector $\mathbf{p}$ into a component $\mathbf{p}_\perp$ perpendicular to the boundary and a component $\mathbf{p}_\parallel$ parallel to the boundary. Let $\Delta U$ be the (signed) difference in potential energy on the two sides of the discontinuity. If $\|\mathbf{p}_\perp\|^2 > 2\Delta U$ then $\mathbf{p}_\perp$ is instantaneously replaced by $\mathbf{p}'_\perp := \sqrt{\|\mathbf{p}_\perp\|^2 - 2\Delta U} \cdot \frac{\mathbf{p}_\perp}{\|\mathbf{p}_\perp\|}$. That is, the discontinuity is passed, but the momentum is changed in the direction perpendicular to the boundary (refraction). (If $\Delta U$ is positive, the momentum will decrease, and if it is negative, the momentum will increase.) On the other hand, if $\|\mathbf{p}_\perp\|^2 \leq 2\Delta U$, then $\mathbf{p}_\perp$ is instantaneously replaced by $-\mathbf{p}_\perp$. That is, if the particle's momentum is insufficient to climb the potential boundary, it bounces back by reversing the momentum component which is perpendicular to the boundary.

Pakman and Paninski [11, 10] present an algorithm to exactly solve these dynamics for quadratic $U$. However, for non-quadratic $U$, the Hamiltonian dynamics rarely have a closed-form solution, and one must resort to numerical integration, the most successful method for which is known as the *leapfrog* dynamics.

# 4 Reflection and Refraction with Leapfrog Dynamics

Informally, HMC with leapfrog dynamics iterates three steps. (1) Sample $\mathbf{p} \sim P(\mathbf{p})$. (2) Perform leapfrog simulation, by discretizing the Hamiltonian equations into $L$ steps using some small step-size $\epsilon$. Here, one interleaves a *position step* $\mathbf{q} \leftarrow \mathbf{q} + \epsilon \mathbf{p}$ between two half *momentum* steps $\mathbf{p} \leftarrow \mathbf{p} - \epsilon \nabla U(\mathbf{q})/2$. (3) Reverse the sign of $\mathbf{p}$. If $(\mathbf{q}, \mathbf{p})$ is the starting point of the leapfrog dynamics, and $(\mathbf{q}', \mathbf{p}')$ is the final point, *accept* the move with probability $\min(1, \exp(H(\mathbf{p}, \mathbf{q}) - H(\mathbf{p}', \mathbf{q}')))$ See Algorithm 1.

It can be shown that this baseline HMC method has detailed balance with respect to $P(\mathbf{p})$, even if $U(\mathbf{q})$ is discontinuous. However, discontinuities mean that large changes in the Hamiltonian may occur, meaning many steps can be rejected. We propose a modification of the dynamics, namely, *reflective Hamiltonian Monte Carlo* (RHMC), which is also shown in Algorithm 1.

The only modification is applied to the position steps: In RHMC, the first intersection of the trajectory with the boundaries of the polytope that contains $\mathbf{q}$ must be detected [11, 10]. The position step is only taken up to this boundary, and reflection/refraction occurs, depending on the momentum and change of energy at the boundary. This process continues until the entire amount of time $\epsilon$ has been simulated. Note that if there is no boundary in the trajectory to time $\epsilon$, this is equivalent to baseline HMC. Also note that several boundaries might be visited in one position step.

As with baseline HMC, there are two alternating steps, namely drawing a new momentum variable $\mathbf{p}$ from $P(\mathbf{p}) \propto \exp(-K(\mathbf{p}))$ and proposing a move $(\mathbf{p}, \mathbf{q}) \to (\mathbf{p}', \mathbf{q}')$ and accepting or rejecting it with a probability determined by a Metropolis-Hastings ratio. We can show that both of these steps leave the joint distribution $P$ invariant, and hence a Markov chain that also alternates between these steps will also leave $P$ invariant.

As it is easy to see, drawing $\mathbf{p}$ from $P(\mathbf{p})$ will leave $P(\mathbf{q}, \mathbf{p})$ invariant, we concentrate on the second step i.e. where a move is proposed according to the piecewise leapfrog dynamics shown in Alg. 1. Firstly, it is clear that these dynamics are time-reversible, meaning that if the simulation takes state $(\mathbf{q}, \mathbf{p})$ to $(\mathbf{q}', \mathbf{p}')$ it will also take state $(\mathbf{q}', \mathbf{p}')$ to $(\mathbf{q}, \mathbf{p})$. Secondly, we will show that these dynamics are volume preserving. Formally, if $\mathcal{D}$ denotes the leapfrog dynamics, we will show that the absolute value of the determinant of the Jacobian of $\mathcal{D}$ is one. These two properties together show that the probability density of proposing a move from $(\mathbf{q}, \mathbf{p})$ to $(\mathbf{q}', \mathbf{p}')$ is the same of that proposing a move from $(\mathbf{q}', \mathbf{p}')$ to $(\mathbf{q}, \mathbf{p})$. Thus, if the move $(\mathbf{q}, \mathbf{p}) \to (\mathbf{q}', \mathbf{p}')$ is accepted according to the standard Metropolis-Hastings ratio, $R((\mathbf{q}, \mathbf{p}) \to (\mathbf{q}', \mathbf{p}')) = \min(1, \exp(H(\mathbf{q}, \mathbf{p}) - H(\mathbf{q}', \mathbf{p}'))$, then detailed balance will be satisfied. To see this, let $Q$ denote the proposal distribution, Then, the usual proof of correctness for Metropolis-Hastings applies, namely that

$$\frac{P(\mathbf{q},\mathbf{p})Q\left((\mathbf{q},\mathbf{p})\to(\mathbf{q}',\mathbf{p}')\right)R\left((\mathbf{q},\mathbf{p})\to(\mathbf{q}',\mathbf{p}')\right)}{P(\mathbf{q}',\mathbf{p}')Q((\mathbf{q}',\mathbf{p}')\to(\mathbf{q},\mathbf{p}))R((\mathbf{q}',\mathbf{p}')\to(\mathbf{q},\mathbf{p}))}$$

$$= \frac{P(\mathbf{q},\mathbf{p})\min(1,\exp(H(\mathbf{q},\mathbf{p})-H(\mathbf{q}',\mathbf{p}')))}{P(\mathbf{q}',\mathbf{p}')\min(1,\exp(H(\mathbf{q}',\mathbf{p}')-H(\mathbf{q},\mathbf{p})))}=1. \quad (3)$$

(The final equality is easy to establish, considering the cases where $H(\mathbf{q},\mathbf{p}) \geq H(\mathbf{q}',\mathbf{p}')$ and $H(\mathbf{q}',\mathbf{p}') \leq H(\mathbf{q},\mathbf{p})$ separately.) This means that detailed balance holds, and so $P$ is a stationary distribution.

The major difference in the analysis of RHMC, relative to traditional HMC is that showing conservation of volume is more difficult. With standard HMC and leapfrog steps, volume conservation is easy to show by observing that each part of a leapfrog step is a shear transformation. This is not the case with RHMC, and so we must resort to a full analysis of the determinant of the Jacobian, as explored in the following section.

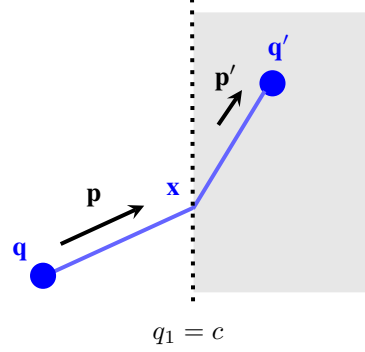

$$q_1 = c$$

Figure 2: Transformation $\mathcal{T}:\langle\mathbf{q},\mathbf{p}\rangle\to\langle\mathbf{q}',\mathbf{p}'\rangle$ described by Lemma 1 (Refraction).

---

**Algorithm 1**: BASELINE & REFLECTIVE HMC ALGORITHMS

    **input** : $\mathbf{q}_0$, current sample; $U$, potential function, $L$, # leapfrog steps; $\epsilon$, leapfrog step size
    **output**: next sample

1  **begin**
2     $\mathbf{q}\leftarrow\mathbf{q}_0;\quad \mathbf{p}\sim\mathcal{N}(0,1)$
3     $H_0\leftarrow\|\mathbf{p}\|^2/2+U(\mathbf{q})$
4     **for** $l=1$ **to** $L$ **do**
5         $\mathbf{p}\leftarrow\mathbf{p}-\epsilon\nabla U(\mathbf{q})/2$                     *# Half-step evolution of momentum*
6         **if** BASELINEHMC **then**               *# Full-step evolution of position*:
7           $\mathbf{q}\leftarrow\mathbf{q}+\epsilon\mathbf{p}$
8         **else**                            *# i.e. if* REFLECTIVEHMC:
9           $t_0\leftarrow 0$
10          **while** $\left(\langle\boldsymbol{x},t_x,\Delta U,\phi\rangle\leftarrow\text{FIRSTDISCONTINUITY}(\boldsymbol{q},\boldsymbol{p},\epsilon-t_0,U)\right)\neq\emptyset$ **do**
11             $\mathbf{q}\leftarrow\mathbf{x}$
12             $t_0\leftarrow t_0+t_x$
13             $\langle\mathbf{p}_\perp,\mathbf{p}_\|\rangle=\text{DECOMPOSE}(\mathbf{p},\phi)$     *# Perpendicular/ parallel to boundary plane $\phi$*
14             **if** $\|\boldsymbol{p}_\perp\|^2>2\Delta U$ **then**
15                $\mathbf{p}_\perp\leftarrow\sqrt{\|\mathbf{p}_\perp\|^2-2\Delta U}\cdot\frac{\mathbf{p}_\perp}{\|\mathbf{p}_\perp\|}$         *# Refraction*
16             **else**
17                $\mathbf{p}_\perp\leftarrow-\mathbf{p}_\perp$                  *# Reflection*
18
19             $\mathbf{p}\leftarrow\mathbf{p}_\perp+\mathbf{p}_\|$
20          $\mathbf{q}\leftarrow\mathbf{q}+(\epsilon-t_0)\mathbf{p}$
21         $\mathbf{p}\leftarrow\mathbf{p}-\epsilon\nabla U(\mathbf{q})/2$                 *# Half-step evolution of momentum*
22     $\mathbf{p}\leftarrow-\mathbf{p}$               *# Not required in practice; for reversibility proof*
23     $H\leftarrow\|\mathbf{p}\|^2/2+U(\mathbf{q});\quad \Delta H\leftarrow H-H_0$
24     **if** $s\sim\mathcal{U}(0,1)<e^{-\Delta H}$ **return q else return** $\mathbf{q}_0$
25 **end**

    **note** : FIRSTDISCONTINUITY$(\cdot)$ returns **x**, the position of the first intersection of a boundary plain with line segment $[\mathbf{q},\mathbf{q}+(\epsilon-t_0)\mathbf{p}]$; $t_x$, the time it is visited; $\Delta U$, the change in energy at the discontinuity, and $\phi$, the visited partition boundary. If no such point exists, $\emptyset$ is returned.

# 5 Volume Conservation

## 5.1 Refraction

In our first result, we assume without loss of generality, that there is a boundary located at the hyperplane $q_1 = c$. This Lemma shows that, in the refractive case, volume is conserved. The setting is visualized in Figure 2.

**Lemma 1.** *Let $\mathcal{T} : \langle \boldsymbol{q}, \boldsymbol{p} \rangle \to \langle \boldsymbol{q}', \boldsymbol{p}' \rangle$ be a transformation in $\mathbb{R}^n$ that takes a unit mass located at $\boldsymbol{q} := (q_1, \ldots, q_n)$ and moves it with constant momentum $\boldsymbol{p} := (p_1, \ldots, p_n)$ till it reaches a plane $q_1 = c$ (at some point $\boldsymbol{x} := (c, x_2, \ldots, x_n)$ where $c$ is a constant). Subsequently the momentum is changed to $\boldsymbol{p}' = \left( \sqrt{p_1^2 - 2\Delta U(\boldsymbol{x})}, p_2, \ldots, p_n \right)$ (where $\Delta U(\cdot)$ is a function of $\boldsymbol{x}$ s.t. $p_1^2 > 2\Delta U(\boldsymbol{x})$). The move is carried on for the total time period $\tau$ till it ends in $\boldsymbol{q}'$. For all $n \in \mathbb{N}$, $\mathcal{T}$ satisfies the volume preservation property.*

*Proof.* Since for $i > 1$, the momentum is not affected by the collision, $q_i' = q_i + \tau \cdot p_i$ and $p_i' = p_i$. Thus,

$$\forall j \in \{2, \ldots, n\} \, s.t. \, j \neq i, \qquad \frac{\partial q_i'}{\partial q_j} = \frac{\partial q_i'}{\partial p_j} = \frac{\partial p_i'}{\partial q_j} = \frac{\partial p_i'}{\partial p_j} = 0.$$

Therefore, if we explicitly write out the Jacobian determinant $|J|$ of the transformation $\mathcal{T}$, it is

$$
\begin{vmatrix}
\frac{\partial q_1'}{\partial q_1} & \frac{\partial q_1'}{\partial p_1} & \cdots & \frac{\partial q_1'}{\partial p_{k-1}} & \frac{\partial q_1'}{\partial q_k} & \frac{\partial q_1'}{\partial p_k} \\
\frac{\partial p_1'}{\partial q_1} & \frac{\partial p_1'}{\partial p_1} & \cdots & \frac{\partial p_1'}{\partial p_{k-1}} & \frac{\partial p_1'}{\partial q_k} & \frac{\partial p_1'}{\partial p_k} \\
\frac{\partial q_2'}{\partial q_1} & \frac{\partial q_2'}{\partial p_1} & \cdots & \frac{\partial q_2'}{\partial p_{k-1}} & \frac{\partial q_2'}{\partial q_k} & \frac{\partial q_2'}{\partial p_k} \\
\vdots & & \ddots & \vdots & & \vdots \\
\frac{\partial p_{k-1}'}{\partial q_1} & \frac{\partial p_{k-1}'}{\partial p_1} & \cdots & \frac{\partial p_{k-1}'}{\partial p_{k-1}} & \frac{\partial p_{k-1}'}{\partial q_k} & \frac{\partial p_{k-1}'}{\partial p_k} \\
\frac{\partial q_k'}{\partial q_1} & \frac{\partial q_k'}{\partial p_1} & \cdots & \frac{\partial q_k'}{\partial p_{k-1}} & \frac{\partial q_k'}{\partial q_k} & \frac{\partial q_k'}{\partial p_k} \\
\frac{\partial p_k'}{\partial q_1} & \frac{\partial p_k'}{\partial p_1} & \cdots & \frac{\partial p_k'}{\partial p_{k-1}} & \frac{\partial p_k'}{\partial q_k} & \frac{\partial p_k'}{\partial p_k}
\end{vmatrix}
=
\begin{vmatrix}
\frac{\partial q_1'}{\partial q_1} & \frac{\partial q_1'}{\partial p_1} & \cdots & \frac{\partial q_1'}{\partial p_{k-1}} & \frac{\partial q_1'}{\partial q_k} & \frac{\partial q_1'}{\partial p_k} \\
\frac{\partial p_1'}{\partial q_1} & \frac{\partial p_1'}{\partial p_1} & \cdots & \frac{\partial p_1'}{\partial p_{k-1}} & \frac{\partial p_1'}{\partial q_k} & \frac{\partial p_1'}{\partial p_k} \\
0 & 0 & \cdots & \frac{\partial q_2'}{\partial p_{k-1}} & \frac{\partial q_2'}{\partial q_k} & \frac{\partial q_2'}{\partial p_k} \\
\vdots & & \ddots & \vdots & & \vdots \\
0 & 0 & \cdots & 1 & \frac{\partial p_{k-1}'}{\partial q_k} & \frac{\partial p_{k-1}'}{\partial p_k} \\
0 & 0 & \cdots & 0 & 1 & \frac{\partial q_k'}{\partial p_k} \\
0 & 0 & \cdots & 0 & 0 & 1
\end{vmatrix}
\tag{4}
$$

Now, using standard properties of the determinant, we have that $|J| = \begin{vmatrix} \frac{\partial q_1'}{\partial q_1} & \frac{\partial q_1'}{\partial p_1} \\ \frac{\partial p_1'}{\partial q_1} & \frac{\partial p_1'}{\partial p_1} \end{vmatrix}$.

We will now explicitly calculate these four derivatives. Due to the significance of the result, we carry out the computations in detail. Nonetheless, as this is a largely mechanical process, for brevity, we do not comment on the derivation.

Let $t_1$ be the time to reach $\mathbf{x}$ and $t_2$ be the period between reaching $\mathbf{x}$ and the last point $\mathbf{q}'$. Then:

$$t_1 \stackrel{\text{def}}{=} \frac{c - q_1}{p_1} \quad (5) \qquad\qquad \mathbf{x} = \mathbf{q} + t_1 \mathbf{p} \quad (6) \qquad\qquad t_2 \stackrel{\text{def}}{=} \tau - t_1 = \tau + \frac{q_1 - c}{p_1} \quad (7)$$

$$q_1' = c + p_1' \cdot t_2 \quad (8) \qquad p_1' \stackrel{\text{def}}{=} \sqrt{p_1^2 - 2\Delta U(\mathbf{x})} \quad (9) \qquad\qquad \frac{\partial t_2}{\partial q_1} \stackrel{\text{by (7)}}{=} \frac{1}{p_1} \quad (10)$$

$$\frac{\partial q_1'}{\partial q_1} = \frac{\partial q_1'}{\partial p_1'} \cdot \frac{\partial p_1'}{\partial q_1} + \frac{\partial q_1'}{\partial t_2} \cdot \frac{\partial t_2}{\partial q_1} \stackrel{\text{(8 \& 10)}}{=} t_2 \cdot \frac{\partial p_1'}{\partial q_1} + p_1' \cdot \frac{1}{p_1} \tag{11}$$

$$\frac{\partial q_1'}{\partial p_1} = \frac{\partial q_1'}{\partial p_1'} \cdot \frac{\partial p_1'}{\partial p_1} + \frac{\partial q_1'}{\partial t_2} \cdot \frac{\partial t_2}{\partial p_1} \stackrel{\text{(7 \& 8)}}{=} t_2 \cdot \frac{\partial p_1'}{\partial p_1} + p_1' \cdot \frac{c - q_1}{p_1^2} \tag{12}$$

$$\frac{\partial p_1'}{\partial p_1} \stackrel{\text{(9)}}{=} \frac{1}{2\sqrt{p_1^2 - 2\Delta U(\mathbf{x})}} \cdot \frac{\partial \left( p_1^2 - 2\Delta U(\mathbf{x}) \right)}{\partial p_1} = \frac{p_1 - \partial \Delta U(\mathbf{x}) / \partial p_1}{p_1'} \tag{13}$$

$$\frac{\partial p_1'}{\partial q_1} = \frac{1}{2\sqrt{p_1^2 - 2\Delta U(\mathbf{x})}} \cdot \frac{\partial\left(p_1^2 - 2\Delta U(\mathbf{x})\right)}{\partial q_1} = \frac{1}{p_1'} \cdot \frac{-\partial\Delta U(\mathbf{x})}{\partial q_1} \tag{14}$$

$$\frac{\partial\mathbf{x}}{\partial p_1} \overset{(5,\,6)}{=} \frac{\partial\left(\mathbf{q} + \frac{c-q_1}{p_1}\mathbf{p}\right)}{\partial p_1} = \frac{\partial\mathbf{q}}{\partial p_1} + (c - q_1)\cdot\frac{\partial(\mathbf{p}/p_1)}{\partial p_1} = (c - q_1)\frac{-1}{p_1^2}\cdot(0, p_2, p_3, \ldots, p_n) \tag{15}$$

$$\frac{\partial\mathbf{x}}{\partial q_1} \overset{(5,\,6)}{=} \frac{\partial\mathbf{q}}{\partial q_1} + \frac{\mathbf{p}}{p_1}\cdot\frac{\partial(c - q_1)}{\partial q_1} = \frac{\mathbf{q}}{q_1} - \frac{\mathbf{p}}{p_1} = (1, 0, \ldots, 0) - (1, \frac{p_2}{p_1}, \ldots, \frac{p_n}{p_1}) = \frac{-1}{p_1}(0, p_2, \ldots, p_n) \tag{16}$$

Substituting the appropriate terms into $|J| = |\frac{\partial q_1'}{\partial q_1}\frac{\partial p_1'}{\partial p_1} - \frac{\partial p_1'}{\partial q_1}\frac{\partial q_1'}{\partial p_1}|$, we get that

$$
\begin{aligned}
|J| &\overset{(4)}{=} \frac{\partial q_1'}{\partial q_1}\cdot\frac{\partial p_1'}{\partial p_1} - \frac{\partial q_1'}{\partial p_1}\cdot\frac{\partial p_1'}{\partial q_1} \overset{(11\,\&\,12)}{=} \left(t_2\frac{\partial p_1'}{\partial q_1} + \frac{p_1'}{p_1}\right)\cdot\frac{\partial p_1'}{\partial p_1} - \left(t_2\frac{\partial p_1'}{\partial p_1} + p_1'\frac{c - q_1}{p_1^2}\right)\cdot\frac{\partial p_1'}{\partial q_1} \\
&= \frac{p_1'}{p_1}\left(\frac{\partial p_1'}{\partial p_1} + \frac{q_1 - c}{p_1}\cdot\frac{\partial p_1'}{\partial q_1}\right) \overset{(13\,\&\,14)}{=} \frac{1}{p_1}\left(p_1 - \frac{\partial\Delta U(\mathbf{x})}{\partial p_1} - \frac{q_1 - c}{p_1}\cdot\frac{\partial\Delta U(\mathbf{x})}{\partial q_1}\right) \\
&= 1 - \frac{1}{p_1}\left(\frac{\partial\Delta U(\mathbf{x})}{\partial\mathbf{x}}\cdot\frac{\partial\mathbf{x}}{\partial p_1} + \frac{q_1 - c}{p_1}\cdot\frac{\partial\Delta U(\mathbf{x})}{\partial\mathbf{x}}\cdot\frac{\partial\mathbf{x}}{\partial q_1}\right) \\
&\overset{(15\,\&\,16)}{=} 1 - \frac{1}{p_1}\cdot\frac{\partial\Delta U(\mathbf{x})}{\partial\mathbf{x}}\left(\frac{q_1 - c}{p_1^2}\cdot(0, p_2, p_3, \ldots, p_n) + \frac{q_1 - c}{p_1}\cdot\frac{-1}{p_1}(0, p_2, \ldots, p_n)\right) = 1.
\end{aligned}
$$

$\square$

## 5.2 Reflection

Now, we turn to the reflective case, and again show that volume is conserved. Again, we assume without loss of generality that there is a boundary located at the hyperplane $q_1 = c$.

**Lemma 2.** *Let $\mathcal{T} : \langle \mathbf{q}, \mathbf{p}\rangle \to \langle \mathbf{q}', \mathbf{p}'\rangle$ be a transformation in $\mathbb{R}^n$ that takes a unit mass located at $\mathbf{q} := (q_1, \ldots, q_n)$ and moves it with the constant momentum $\mathbf{p} := (p_1, \ldots, p_n)$ till it reaches a plane $q_1 = c$ (at some point $\mathbf{x} := (c, x_2, \ldots, x_n)$ where $c$ is a constant). Subsequently the mass is bounced back (reflected) with momentum $\mathbf{p}' = (-p_1, p_2, \ldots, p_n)$ The move is carried on for a total time period $\tau$ till it ends in $\mathbf{q}'$. For all $n \in \mathbb{N}$, $\mathcal{T}$ satisfies the volume preservation property.*

*Proof.* Similar to Lemma 1, for $i > 1$, $q_i' = q_i + \tau\cdot p_i$ and $p_i' = p_i$. Therefore, for any $j \in \{2, \ldots, n\}$ s.t. $j \neq i$, $\quad \frac{\partial q_i'}{\partial q_j} = \frac{\partial q_i'}{\partial p_j} = \frac{\partial p_i'}{\partial q_j} = \frac{\partial p_i'}{\partial p_j} = 0$. Consequently, by equation (4), and since $p_1' = -p_1$,

$$|J| = \begin{vmatrix} \frac{\partial q_1'}{\partial q_1} & \frac{\partial q_1'}{\partial p_1} \\ \frac{\partial p_1'}{\partial q_1} & \frac{\partial p_1'}{\partial p_1} \end{vmatrix} = \begin{vmatrix} \frac{\partial q_1'}{\partial q_1} & \frac{\partial q_1'}{\partial p_1} \\ 0 & -1 \end{vmatrix} = \frac{-\partial q_1'}{\partial q_1} \tag{17}$$

As before, let $t_1$ be the time to reach $\mathbf{x}$ and $t_2$ be the period between reaching $\mathbf{x}$ and the last point $\mathbf{q}'$. That is, $t_1 \overset{\text{def}}{=} \frac{c - q_1}{p_1}$ and $t_2 \overset{\text{def}}{=} \tau - t_1$. It follows that $q_1' \overset{\text{def}}{=} c + p_1'\cdot t_2$ is equal to $2c - \tau p_1 - q_1$. Hence, $|J| = 1$. $\square$

## 5.3 Reflective Leapfrog Dynamics

**Theorem 1.** *In RHMC (Algorithm 1) for sampling from a continuous and piecewise distribution $P$ which has affine partitioning boundaries, leapfrog simulation preserves volume in $(\mathbf{q}, \mathbf{p})$ space.*

*Proof.* We split the algorithm into several atomic transformations $\mathcal{T}_i$. Each transformation is either (a) a momentum step, (b) a full position step with no reflection/refraction or (c) a full or partial position step where exactly one reflection or refraction occurs.

To prove that the total algorithm preserves volume, it is sufficient to show that the volume is preserved under each $\mathcal{T}_i$ (i.e. $|J_{\mathcal{T}_i}(\mathbf{q}, \mathbf{p})| = 1$) since:

$$|J_{\mathcal{T}_1 o \mathcal{T}_2 o \cdots o \mathcal{T}_m}| = |J_{\mathcal{T}_1}| \cdot |J_{\mathcal{T}_2}| \cdots |J_{\mathcal{T}_m}|$$

Transformations of kind (a) and (b) are shear mappings and therefore they preserve the volume [9]. Now consider a (full or partial) position step where a single refraction occurs. If the reflective plane is in form $q_1 = c$, by lemma 1, the volume preservation property holds. Otherwise, as long as the reflective plane is affine, via a rotation of basis vectors, the problem is reduced to the former case. Since volume is conserved under rotation, in this case the volume is also conserved. With similar reasoning, by lemma 2, reflection on a affine reflective boundary preserves volume. Thus, since all component transformations of RHMC leapfrog simulation preserve volume, the proof is complete. □

Along with the fact that the leapfrog dynamics are time-reversible, this shows that the algorithm satisfies detailed balance, and so has the correct stationary distribution.

## 6   Experiment

Compared to baseline HMC, we expect that RHMC will simulate Hamiltonian dynamics more accurately and therefore leads to fewer rejected samples. On the other hand, this comes at the expense of slower leapfrog position steps since intersections, reflections and refractions must be computed. To test the trade off, we compare the RHMC to baseline HMC [9] and tuned Metroplis-Hastings (MH) with a simple isotropic Normal proposal distribution. MH is automatically tuned after [12] by testing 100 equidistant proposal variances in interval $(0, 1]$ and accepting a variance for which the acceptance rate is closest to 0.24. The baseline HMC and RHMC number of steps $L$ and step size $\epsilon$ are chosen to be 100 and 0.1 respectively. (Many other choices are in the Appendix.) While HMC performance is highly standard to these parameters [7] RHMC is consistently faster.

The comparison takes place on a heavy tail piecewise model with (non-normalized) negative log probability

$$U(\mathbf{q}) = \begin{cases} \sqrt{\mathbf{q}^\top A \, \mathbf{q}} & \text{if } \|\mathbf{q}\|_\infty \leq 3 \\ 1 + \sqrt{\mathbf{q}^\top A \, \mathbf{q}} & \text{if } 3 < \|\mathbf{q}\|_\infty \leq 6 \\ +\infty, & \text{otherwise} \end{cases} \tag{18}$$

where $A$ is a positive definite matrix. We carry out the experiment on three choices of (position space) dimensionalites, $n = 2, 10$ and $50$.

Due to the symmetry of the model, the ground truth expected value of $\mathbf{q}$ is known to be $\mathbf{0}$. Therefore, the absolute error of the expected value (estimated by a chain $\mathbf{q}^{(1)}, \ldots, \mathbf{q}^{(k)}$ of MCMC samples) in each dimension $d = 1, \ldots, n$ is the absolute value of the mean of $d$-th element of the sample vectors. The worst mean absolute error (WMAE) over all dimensions is taken as the error measurement of the chain.

$$\text{WMAE}\left(\mathbf{q}^{(1)}, \ldots, \mathbf{q}^{(k)}\right) = \frac{1}{k} \max_{d=1,\ldots n} \left| \sum_{s=1}^{k} q_d^s \right| \tag{19}$$

For each algorithm, 20 Markov chains are run and the mean WMAE and 99% confidence intervals (as error bars) versus the number of iterations (i.e. Markov chain sizes) are time (milliseconds) are depicted in figure 2. All algorithms are implemented in java and run on a single thread of a 3.40GHz CPU.

For each of the 20 repetitions, some random starting point is chosen uniformly and used for all three of the algorithms. We use a diagonal matrix for $A$ where, for each repetition, each entry on the main diagonal is either $\exp(-5)$ or $\exp(5)$ with equal probabilities.

As the results show, even in low dimensions, the extra cost of the position step is more or less compensated by its higher effective sample size but as the dimensionality increases, the RHMC significantly outperforms both baseline HMC and tuned MH.

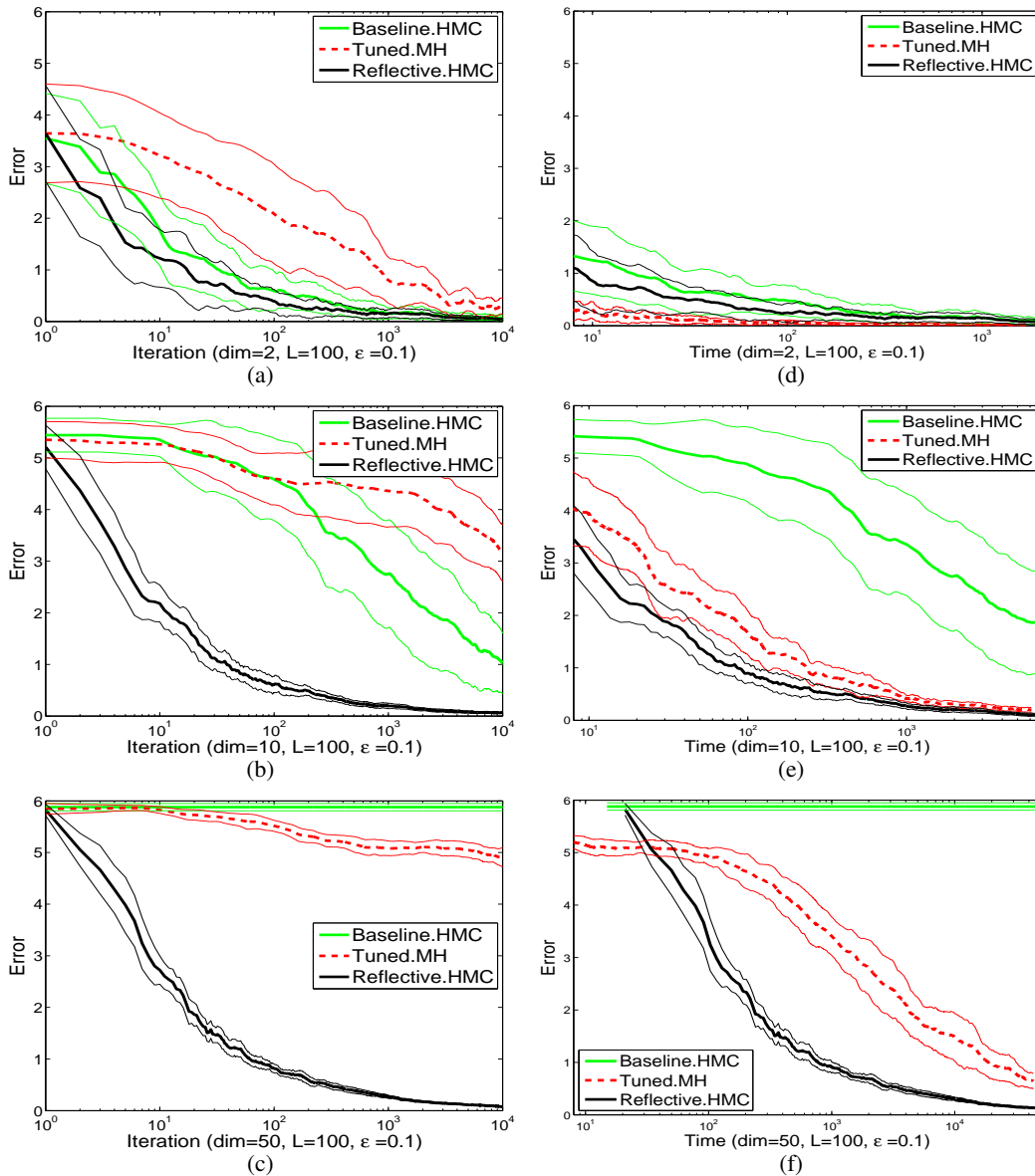

Figure 3: Error (worst mean absolute error per dimension) versus (a-c) iterations and (e-f) time (ms). Tuned.HM is Metropolis Hastings with a tuned isotropic Gaussian proposal distribution. (Many more examples in supplementary material.)

## 7 Conclusion

We have presented a modification of the leapfrog dynamics for Hamiltonian Monte Carlo for piece-wise smooth energy functions with affine boundaries (i.e. each region is a polytope), inspired by physical systems. Though traditional Hamiltonian Monte Carlo can in principle be used on such functions, the fact that the Hamiltonian will often be dramatically changed by the dynamics can result in a very low acceptance ratio, particularly in high dimensions. By better preserving the Hamiltonian, *reflective Hamiltonian Monte Carlo* (RHMC) accepts more moves and thus has a higher effective sample size, leading to much more efficient probabilistic inference. To use this method, one must be able to detect the first intersection of a position trajectory with polytope boundaries.

#### Acknowledgements

NICTA is funded by the Australian Government through the Department of Communications and the Australian Research Council through the ICT Centre of Excellence Program.

## Footnotes

[1]Technically, here we assume the total measure of the non-differentiable points is zero so that, with probability one, none is ever encountered

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
