[Supplementary Material]

# Reflection, Refraction, and Hamiltonian Monte Carlo
# SUPPLEMENTARY MATERIAL

**Hadi Mohasel Afshar**
Research School of Computer Science
Australian National University
Canberra, ACT 0200
hadi.afshar@anu.edu.au

**Justin Domke**
National ICT Australia (NICTA) &
Australian National University
Canberra, ACT 0200
Justin.Domke@nicta.com.au

## 1  Particle trajectories

To visualize the mechanism of the proposed reflective HMC (RHMC), in this section we provide more example trajectories on the test model studied in the paper that is,

$$U(\mathbf{q}) = \begin{cases} \sqrt{\mathbf{q}^\top A\,\mathbf{q}} & \text{if } \|\mathbf{q}\|_\infty \leq 3 \\ 1 + \sqrt{\mathbf{q}^\top A\,\mathbf{q}} & \text{if } 3 < \|\mathbf{q}\|_\infty \leq 6 \\ +\infty, & \text{otherwise} \end{cases}$$

Figure 1 depicts the contours of a two dimensional model where entries on the main dimension of the matrix $A$ are $\exp(-5)$. The proposal trajectories generated by baseline HMC as well as its reflective counterpart are depicted in Figures 2 to 4.

Figure 1: (a) Contours of the demonstration energy function $E(x)$ in two dimensions. (b) Contours of the target distribution $p(x) \propto \exp(-E(x))$.

Figure 2: Example trajectories of baseline and reflective HMC with $L = 25$ and $\epsilon = 0.5$ red and blue lines show rejected and accepted proposals, respectively. (a) Baseline HMC, (b) Reflective HMC.

Figure 3: Example trajectories of baseline and reflective HMC with $L = 25$ and $\epsilon = 0.1$ red and blue lines show rejected and accepted proposals, respectively. (a) Baseline HMC, (b) Reflective HMC.

Figure 4: Example trajectories of baseline and reflective HMC with $L = 25$ and $\epsilon = 0.2$ red and blue lines show rejected and accepted proposals, respectively. (a) Baseline HMC, (b) Reflective HMC.

## 2 Error vs. iteration and time plots

To study the effect of tuning on the baseline and reflective HMC algorithms, in this section we repeat the experiment of the main paper using different leap frog parameter configurations. Namely, in the following figures *error vs iteration* and *error vs time* plots, all combinations of $L \in \{20, 50, 100, 200\}$ and $\epsilon \in \{0.05, 0.1, 0.2, 0.4\}$ are tested on 2, 10 and 50 dimensional models. The worst mean absolute error (WMAE) over all dimensions is taken as the error measurement of a Markov chain. Each experiment is repeated over 10 Markov chains and the means and standard errors are depicted.

These figures show that

1. The proposed reflective HMC (RHMC) remains superior to the baseline HMC over a large range of $L$ and $\epsilon$ and in no setting it performs worse.

2. Compared to the basic HMC, the performance of reflective HMC is less sensitive to parameter tuning.

3. Regardless of the tuning, in high dimensional models, reflective HMC performs significantly better than the basic HMC and tuned Metropolis-Hastings.

Figure 5: Error (worst mean absolute error per dimension) versus iterations (left) and time (right) (ms). Results correspond leapfrog parameters $L = 20$, $\epsilon = 0.05$. Here, Tuned.HM is Metropolis Hastings with a isotropic Gaussian proposal distribution, tuned as described in the main paper.

Figure 6: Error (worst mean absolute error per dimension) versus iterations (left) and time (right) (ms). Results correspond leapfrog parameters $L = 20$, $\epsilon = 0.1$. Here, Tuned.HM is Metropolis Hastings with a isotropic Gaussian proposal distribution, tuned as described in the main paper.

Figure 7: Error (worst mean absolute error per dimension) versus iterations (left) and time (right) (ms). Results correspond leapfrog parameters $L = 20$, $\epsilon = 0.2$. Here, Tuned.HM is Metropolis Hastings with a isotropic Gaussian proposal distribution, tuned as described in the main paper.

Figure 8: Error (worst mean absolute error per dimension) versus iterations (left) and time (right) (ms). Results correspond leapfrog parameters $L = 20$, $\epsilon = 0.4$. Here, Tuned.HM is Metropolis Hastings with a isotropic Gaussian proposal distribution, tuned as described in the main paper.

Figure 9: Error (worst mean absolute error per dimension) versus iterations (left) and time (right) (ms). Results correspond leapfrog parameters $L = 50$, $\epsilon = 0.05$. Here, Tuned.HM is Metropolis Hastings with a isotropic Gaussian proposal distribution, tuned as described in the main paper.

Figure 10: Error (worst mean absolute error per dimension) versus iterations (left) and time (right) (ms). Results correspond leapfrog parameters $L = 50$, $\epsilon = 0.1$. Here, Tuned.HM is Metropolis Hastings with a isotropic Gaussian proposal distribution, tuned as described in the main paper.

Figure 11: Error (worst mean absolute error per dimension) versus iterations (left) and time (right) (ms). Results correspond leapfrog parameters $L = 50$, $\epsilon = 0.2$. Here, Tuned.HM is Metropolis Hastings with a isotropic Gaussian proposal distribution, tuned as described in the main paper.

Figure 12: Error (worst mean absolute error per dimension) versus iterations (left) and time (right) (ms). Results correspond leapfrog parameters $L = 50$, $\epsilon = 0.4$. Here, Tuned.HM is Metropolis Hastings with a isotropic Gaussian proposal distribution, tuned as described in the main paper.

Figure 13: Error (worst mean absolute error per dimension) versus iterations (left) and time (right) (ms). Results correspond leapfrog parameters $L = 100$, $\epsilon = 0.05$. Here, Tuned.HM is Metropolis Hastings with a isotropic Gaussian proposal distribution, tuned as described in the main paper.

Figure 14: Error (worst mean absolute error per dimension) versus iterations (left) and time (right) (ms). Results correspond leapfrog parameters $L = 100$, $\epsilon = 0.1$. Here, Tuned.HM is Metropolis Hastings with a isotropic Gaussian proposal distribution, tuned as described in the main paper.

Figure 15: Error (worst mean absolute error per dimension) versus iterations (left) and time (right) (ms). Results correspond leapfrog parameters $L = 100$, $\epsilon = 0.2$. Here, Tuned.HM is Metropolis Hastings with a isotropic Gaussian proposal distribution, tuned as described in the main paper.

Figure 16: Error (worst mean absolute error per dimension) versus iterations (left) and time (right) (ms). Results correspond leapfrog parameters $L = 100$, $\epsilon = 0.4$. Here, Tuned.HM is Metropolis Hastings with a isotropic Gaussian proposal distribution, tuned as described in the main paper.

Figure 17: Error (worst mean absolute error per dimension) versus iterations (left) and time (right) (ms). Results correspond leapfrog parameters $L = 200$, $\epsilon = 0.05$. Here, Tuned.HM is Metropolis Hastings with a isotropic Gaussian proposal distribution, tuned as described in the main paper.

Figure 18: Error (worst mean absolute error per dimension) versus iterations (left) and time (right) (ms). Results correspond leapfrog parameters $L = 200$, $\epsilon = 0.1$. Here, Tuned.HM is Metropolis Hastings with a isotropic Gaussian proposal distribution, tuned as described in the main paper.

Figure 19: Error (worst mean absolute error per dimension) versus iterations (left) and time (right) (ms). Results correspond leapfrog parameters $L = 200$, $\epsilon = 0.2$. Here, Tuned.HM is Metropolis Hastings with a isotropic Gaussian proposal distribution, tuned as described in the main paper.

Figure 20: Error (worst mean absolute error per dimension) versus iterations (left) and time (right) (ms). Results correspond leapfrog parameters $L = 200$, $\epsilon = 0.4$. Here, Tuned.HM is Metropolis Hastings with a isotropic Gaussian proposal distribution, tuned as described in the main paper.

Table 1: Rate of rejections/reflections and refractions for sampling 10,000 particles using baseline HMC (HMC) and reflective HMC (RHMC) where model dimension = 2. (The rates are averaged over 10 Markov chains.)

| L | Alg. | $\epsilon$ | | | |
|---|------|------|-----|-----|-----|
| | | 0.05 | 0.1 | 0.2 | 0.4 |
| 10 | HMC | no. reject: 1492.7 | no. reject: 3479.9 | no. reject: 6087.2 | no. reject: 8358.4 |
| | RHMC | no. reject: 1280.6 | no. reject: 2892.8 | no. reject: 4904.6 | no. reject: 6880.9 |
| | | no. reflect: 298.5 | no. reflect: 839.1 | no. reflect: 2007.2 | no. reflect: 2972.2 |
| | | no. refract: 217.1 | no. refract: 585.3 | no. refract: 1365.1 | no. refract: 10152.1 |
| 20 | HMC | no. reject: 1966.8 | no. reject: 5120.1 | no. reject: 7673.1 | no. reject: 9564.6 |
| | RHMC | no. reject: 1307.7 | no. reject: 4791.3 | no. reject: 6055.1 | no. reject: 8866.2 |
| | | no. reflect: 785.1 | no. reflect: 931.9 | no. reflect: 3758.6 | no. reflect: 13977.5 |
| | | no. refract: 556.4 | no. refract: 734.3 | no. refract: 2603.7 | no. refract: 36627.5 |
| 50 | HMC | no. reject: 3587.7 | no. reject: 6696.4 | no. reject: 9195.2 | no. reject: 9510.6 |
| | RHMC | no. reject: 2012.2 | no. reject: 4510.8 | no. reject: 8175.9 | no. reject: 7958.5 |
| | | no. reflect: 2129.6 | no. reflect: 4685.6 | no. reflect: 8018.5 | no. reflect: 59906.3 |
| | | no. refract: 1567.7 | no. refract: 3165.8 | no. refract: 16627.1 | no. refract: 121706.7 |
| 100 | HMC | no. reject: 5288.0 | no. reject: 8264.3 | no. reject: 8904.9 | no. reject: 9416.3 |
| | RHMC | no. reject: 3152.5 | no. reject: 5495.8 | no. reject: 6510.4 | no. reject: 6998.3 |
| | | no. reflect: 4029.0 | no. reflect: 10549.3 | no. reflect: 20825.6 | no. reflect: 172305.0 |
| | | no. refract: 2920.2 | no. refract: 7266.3 | no. refract: 51209.3 | no. refract: 261964.0 |
| 200 | HMC | no. reject: 7100.1 | no. reject: 8421.4 | no. reject: 9221.1 | no. reject: 9426.0 |
| | RHMC | no. reject: 2366.9 | no. reject: 6450.9 | no. reject: 6679.2 | no. reject: 7047.6 |
| | | no. reflect: 11509.1 | no. reflect: 18628.0 | no. reflect: 66031.6 | no. reflect: 498257.6 |
| | | no. refract: 7693.9 | no. refract: 22783.0 | no. refract: 143159.7 | no. refract: 645459.2 |

## 3  The rate of rejections, reflections and refractions

In this section, for a range of parameter tunings and model dimensions, the proposal rejection rate of the basic and reflective HMC algorithms as well as the number of reflections and refractions of the latter sampler are depicted in Tables 1 to 3. An interesting observation is that in high dimensions (as in Table 3), the number of refractions is significantly less that reflections.

This is due to the fact that the studied models consist of two positive-probability (hyper-cubic) partitions one of which is inside the other. In high dimensions, the internal hyper-cube is rarely sampled since most of the mass is placed near the outer surface of the external one. Based on this observation, we predict that in general, due to the effect of *reflection*, the performance of our proposed method on high dimensional truncated/finite-support model should be remarkable. The reason is that in such high-dimensional models, the density mass is concentrated near the manifolds that mark the external surfaces of the density function and the reflections prevent the leapfrog mechanism to enter the zero-probability ambient space.

Table 2: Rate of rejections/reflections and refractions for sampling 10,000 particles using baseline HMC (HMC) and reflective HMC (RHMC) where model dimension = 10. (The rates are averaged over 10 Markov chains.)

| L | Alg. | $\epsilon$ | | | |
|---|------|------|-----|-----|-----|
| | | 0.05 | 0.1 | 0.2 | 0.4 |
| 10 | HMC | no. reject: 1492.2 | no. reject: 3677.1 | no. reject: 7284.7 | no. reject: 9664.6 |
| | RHMC | no. reject: 143.3<br>no. reflect: 1437.1<br>no. refract: 235.6 | no. reject: 897.7<br>no. reflect: 3642.0<br>no. refract: 433.9 | no. reject: 4175.4<br>no. reflect: 7713.7<br>no. refract: 660.9 | no. reject: 9076.4<br>no. reflect: 13378.9<br>no. refract: 2301.4 |
| 20 | HMC | no. reject: 2754.0 | no. reject: 5472.9 | no. reject: 8873.8 | no. reject: 9946.3 |
| | RHMC | no. reject: 130.7<br>no. reflect: 3089.3<br>no. refract: 554.4 | no. reject: 799.7<br>no. reflect: 6864.7<br>no. refract: 878.3 | no. reject: 4260.2<br>no. reflect: 15013.6<br>no. refract: 1348.5 | no. reject: 9365.8<br>no. reflect: 32340.3<br>no. refract: 5057.8 |
| 50 | HMC | no. reject: 5738.4 | no. reject: 8337.8 | no. reject: 9629.4 | no. reject: 9987.0 |
| | RHMC | no. reject: 141.9<br>no. reflect: 8035.2<br>no. refract: 1315.6 | no. reject: 680.2<br>no. reflect: 15536.1<br>no. refract: 2731.2 | no. reject: 4921.8<br>no. reflect: 36844.7<br>no. refract: 3553.6 | no. reject: 9497.0<br>no. reflect: 113852.2<br>no. refract: 14880.1 |
| 100 | HMC | no. reject: 8139.5 | no. reject: 9771.6 | no. reject: 9968.1 | no. reject: 9999.7 |
| | RHMC | no. reject: 157.2<br>no. reflect: 16291.1<br>no. refract: 2307.4 | no. reject: 960.0<br>no. reflect: 36164.2<br>no. refract: 4260.1 | no. reject: 4359.8<br>no. reflect: 67230.7<br>no. refract: 11126.0 | no. reject: 9640.1<br>no. reflect: 321873.9<br>no. refract: 24149.7 |
| 200 | HMC | no. reject: 9600.1 | no. reject: 9987.0 | no. reject: 9999.9 | no. reject: 10000.0 |
| | RHMC | no. reject: 156.7<br>no. reflect: 32738.1<br>no. refract: 5022.2 | no. reject: 744.1<br>no. reflect: 64447.1<br>no. refract: 9978.9 | no. reject: 5399.3<br>no. reflect: 168753.9<br>no. refract: 14639.0 | no. reject: 9662.6<br>no. reflect: 1022839.6<br>no. refract: 36450.5 |

Table 3: Rate of rejections/reflections and refractions for sampling 10,000 particles using baseline HMC (HMC) and reflective HMC (RHMC) where model dimension = 50. (The rates are averaged over 10 Markov chains.)

| L | Alg. | $\epsilon$ | | | |
|---|------|------------|---|---|---|
| | | 0.05 | 0.1 | 0.2 | 0.4 |
| 10 | HMC | no. reject: 5455.4 | no. reject: 8267.9 | no. reject: 9618.1 | no. reject: 9997.5 |
| | RHMC | no. reject: 69.1<br>no. reflect: 7609.2<br>no. refract: 0.0 | no. reject: 271.3<br>no. reflect: 17121.9<br>no. refract: 0.0 | no. reject: 1074.3<br>no. reflect: 31497.3<br>no. refract: 0.0 | no. reject: 4157.0<br>no. reflect: 66778.1<br>no. refract: 0.0 |
| 20 | HMC | no. reject: 8301.4 | no. reject: 9719.0 | no. reject: 9999.5 | no. reject: 10000.0 |
| | RHMC | no. reject: 64.2<br>no. reflect: 17257.4<br>no. refract: 0.0 | no. reject: 289.8<br>no. reflect: 33841.4<br>no. refract: 0.0 | no. reject: 988.0<br>no. reflect: 66524.4<br>no. refract: 0.2 | no. reject: 4197.3<br>no. reflect: 141453.9<br>no. refract: 0.0 |
| 50 | HMC | no. reject: 9869.9 | no. reject: 10000.0 | no. reject: 10000.0 | no. reject: 10000.0 |
| | RHMC | no. reject: 59.9<br>no. reflect: 40732.3<br>no. refract: 0.0 | no. reject: 233.8<br>no. reflect: 82537.0<br>no. refract: 0.0 | no. reject: 878.4<br>no. reflect: 160209.2<br>no. refract: 0.0 | no. reject: 3921.5<br>no. reflect: 331565.0<br>no. refract: 0.0 |
| 100 | HMC | no. reject: 9998.8 | no. reject: 10000.0 | no. reject: 10000.0 | no. reject: 10000.0 |
| | RHMC | no. reject: 56.9<br>no. reflect: 85937.0<br>no. refract: 0.4 | no. reject: 215.1<br>no. reflect: 157553.7<br>no. refract: 0.2 | no. reject: 960.5<br>no. reflect: 356408.4<br>no. refract: 0.0 | no. reject: 3899.6<br>no. reflect: 665330.2<br>no. refract: 0.0 |
| 200 | HMC | no. reject: 10000.0 | no. reject: 10000.0 | no. reject: 10000.0 | no. reject: 10000.0 |
| | RHMC | no. reject: 60.2<br>no. reflect: 157838.8<br>no. refract: 0.2 | no. reject: 229.8<br>no. reflect: 335667.9<br>no. refract: 0.0 | no. reject: 888.9<br>no. reflect: 667757.6<br>no. refract: 0.2 | no. reject: 3911.3<br>no. reflect: 1325778.9<br>no. refract: 1.6 |