[Reviews · NeurIPS 2015]

Submitted by Assigned_Reviewer_1

SUMMARY

Hamiltonian MCMC methods sample from a probability

distribution by treating its log as a "potential energy"

function over the state space, augmenting the space with extra

"momentum variables" and their associated "kinetic energy",

and evolving the state of the Markov process by integrating

the physical Hamiltonian equations of motion of the system.

Each step of the Markov chain is accomplished by numerically

integrating the Hamiltonian equations forward in time.

However, if the energy function is non-differentiable, the

integral is not well-defined.

The rejection step that is used

to counteract numerical inaccuracies in the integration also

accounts for such non-differentiable regions, but at the cost

of slowing down the mixing rate of the Markov chain.

This

paper suggests physically-inspired "reflections" and

"refractions" of the trajectory of the system that occur

whenever the state crosses a discontinuity in the energy

function.

It applies to target distributions that are

differentiable everywhere except on the boundaries of certain

polytopes; the reflection or refraction occurs whenever the

trajectory of the system crosses such a boundary.

Whether a reflection or refraction occurs depends on if the

momentum component of the state is sufficient to "climb" the

gap in energy.

The authors show that the necessary

volume-preservation properties are satisfied by these

reflection and refraction procedures, so that the Markov

chains converges to the required target distribution.

QUALITY

The presented modification to Hamiltonian MCMC is useful and

the case of piecewise-differentiable target distributions is

quite applicable in practice.

It would, of course, be useful

to have boundaries that are not affine hyperplanes.

CLARITY

The algorithm, its motivation, and the proof of the volume

preservation property is clearly presented.

Some more

(qualitative and/or quantitative) discussion about how the

reflections and refractions affect rejection rates would be

useful.

ORIGINALITY AND SIGNIFICANCE

The idea of applying physically-based reflection and

refraction to Hamiltonian MCMC methods appears to be novel and

useful.

While the present work proves its correctness under

certain conditions, it is an interesting avenue of research to

extend it further.

Summary: A modification of Hamiltonian MCMC is presented to

handle probability distributions that are not differentiable on

the boundaries of polytopes.

The proposed solution is for the

trajectory of the system to be reflected or refracted when it

encounters such a boundary, thereby reducing rejection rates and

improving mixing speed.

Submitted by Assigned_Reviewer_2

This paper presents and verifies (correctness and performance through simulation) a sensible variation on HMC for pdfs with discontinuities or on a bounded domain.

The method relies on simulating refraction and reflection that result from hamiltonian dynamics, so leads to better proposals around discontinuities (and higher acceptance rates => higher ess).

One thing that would be nice is to see an example of a pdf when it's not worth computing intersections.

What types of high dimensional pdfs is this method not suited for?
Summary: This paper presents and verifies

Submitted by Assigned_Reviewer_3

This is an interesting extension of the standard case and the proof of volume conservation presented is simple but important and was lacking in previous works. The writing is clear.

Typos: l.161:

",Then" l.376: "parameters, are tuned"
Summary: This paper extends the leapfrog discretization used in Hamiltonian Monte Carlo to piecewise smooth distributions. The algorithm is correct and is an interesting extension of the smooth case.

Submitted by Assigned_Reviewer_4

UPDATES AFTER REBUTTAL: Thank you for addressing my questions. The extra numerical results in the supplement are a welcome addition to the paper. I think that this is a good paper, but I will keep my score at 7.

ORIGINAL REVIEW: The paper proposes a modified leapfrog integrator for Hamiltonian Monte Carlo for sampling from piecewise continuous probability distributions, with affine discontinuity boundaries. The simulator makes use of reflection and refraction of the simulated trajectory whenever it hits a boundary. Similar ideas have previously been used for simulating from multivariate Gaussian distributions with constraints, but as far as I can tell this is the first application of this approach to "general" distributions (although, I'm not an expert on HMC). The idea is simple but intuitively appealing (most good ideas are simple!) and I think that this method can be very useful for a restricted class of problems. The numerical evaluation is not very extensive and the authors should consider extending it for a later (re)submission. Furthermore, there are several typos that need to be corrected.

Finally, can you comment on the possibility of using non-affine discontinuity boundaries? Apart from the computational issues associated with computing intersections, would it be possible to extend the approach to allow for more general boundaries?

Some detailed comments/questions:

- In section 3: in think that it could be useful to mention that K(p') = K(p) - \Delta U to explain the form of the refracted momentum. - In Equation (3): aren't the upper triangle (excluding the four elements in the upper left corner) also zero? - Line 363: Missing normalisation by k in the definition of WMAE. - Footnote 2 on page 7: I think that it would be good if you could elaborate on the sensitivity of the tuning parameters (L and eps) of your method to convince the reader. Perhaps include additional numerical result in a supplementary material?
Summary: The paper proposes a modified leapfrog integrator for Hamiltonian Monte Carlo for sampling from piecewise continuous probability distributions, with affine discontinuity boundaries. The idea is simple but potentially very useful for a restricted class of problems. The numerical evaluation is not very extensive.

Author Feedback
Author rebuttal: We would like to thank the reviewers for their helpful and constructive comments.

R1>
1. RE More quantitative/qualitative discussions of the effect of reflections and refractions on the rejection rate:

We have provided more numerical results accessible by the following anonymous link:

https://drive.google.com/file/d/0B9SfUIl9GRZHWWRGOWRwTkx1UmM/view?usp=sharing

or in case it does not work, by the following anonymous link:

http://www.keepandshare.com/doc2/85678/supplimentary-pdf-4-7-meg?da=y

In the last pages of the uploaded file there are three tables that represent the rate of reflections and refractions (as long as rejections) with respect to several different tunings. It also contains a brief discussion on the key role that reflection plays in the sampler's performance in high dimensions.

***

R2>
1. We will fix the typos. Thanks for mentioning them.

***

R3>
1. RE more numerical evaluation:

In the anonymous link mentioned in R1.1 we have provided more numerical evaluations (as suggested).

2. RE elaborating on the sensitivity of the tuning parameters (L and eps) of our method.

In the uploaded supplementary material, our method and the baselines are tested on all combinations of parameter L being in {20, 50, 100, 200}; epsilon being in {0.05, 0.1, 0.2, 0.4} and the model dimension be in {2, 10, 50}.
It is already well known that basic HMC is quite sensitive to tunings (see e.g. the paper's citation [7]).
This is verified by the provided supplementary plots. In contrast, reflective HMC is not so: On the tested parameters, our method is never worse than HMC and in most cases (particularly, in high dimensions) it performs considerably better.

3. RE the possibility of using non-affine discontinuity boundaries:

In this case, it can be shown that analogous to curved mirrors, the volume preservation may be violated (due to magnification). However, despite this, we hope that (with some modifications) we can design a reflection/refraction based HMC that even in the mentioned case, converges to the correct stationary distribution. We cannot be sure whether that is possible but do consider the prospects for extending the work beyond affine borders.

4. In Equation (3): aren't the upper triangle (excluding the four elements in the upper left corner) also zero?

Not all of them. For instance, in the 3rd row and 4th column:

(d q'_2) /(dp_2) = tau.

5. Thanks for the other detailed comments/typos. We will implement/fix them.

***

R4>
1. RE "the definition of the time period is not intuitive":

The total leapfrog full-step time is epsilon. We divide it into arbitrary smaller periods,

Epsilon = tau1 + tau2 +...
such that in each period tau_i, at most one reflection or refraction occurs. According to lemmas 1 or 2 (or in case there is no collision, just a shear transformation) in each period, tau_i, the volume is preserved. Therefore it is preserved for the total time, epsilon, also.

2. RE "it is questionable that the total energy is preserved during the refraction":

Consider a collision to a hyperplane q1 = c (since any collision can be transformed to this case via rotation).
In this case, only the momentum element associated with the first dimension changes.
Let p1' and p1'' be such momentum elements just before and after collision.
In order to preserve the total energy we equate the Hamiltonian
(that is, H(q,p) = U(q) + K(p) where k(p) = (p^2)/2) right before and after the collision:

0.5*(p1')^2 = (\Delta U) + 0.5*(p1'')^2

Therefore in order to preserve the total energy, the momentum just after the collision should be:

(p1'') = SQRT{p1' - 2(\Delta U)}

As used in the paper.
We did not provide the proof in the paper since it is already stated in Pakman's papers [10] and [11]. However, we will either mention it in the paper or provide the above proof explicitly.

***

R5>
1. What types of high-dimensional pdfs is this method not suited for?

According to the discussion provided in the anonymous link (mentioned in R1.1), we believe that if the high dimensional density is truncated or has external boundaries, our method is always suitable.
But if the pdf does not have a finite support (i.e. is not surrounded by a 0-probability space) and it has a large enough number of internal boundaries and the potential gaps at the boundaries are sufficiently small, It would eventually be better off just ignoring them using regular HMC.